

# Exploring the mechanisms by which camel lactoferrin can kill *Salmonella enterica* serovar *typhimurium* and *Shigella sonnei*

Hussein A. Almehdar[1], Nawal Abd El-Baky[2], Ehab H. Mattar[1], Raed Albiheyri[1], Atif Bamagoos[1], Abdullah Aljaddawi[1], Vladimir N. Uversky[1,3] and Elrashdy M. Redwan[1,2]

[1] Department of Biological Sciences, Faculty of Sciences, King Abdulaziz University, Jeddah, Saudi Arabia
[2] Therapeutic and Protective Proteins Laboratory, Protein Research Department, Genetic Engineering and Biotechnology Research Institute, City of Scientific Research and Technological Applications, Alexandria, Egypt
[3] Department of Molecular Medicine, Morsani College of Medicine, University of South Florida, Tampa, FL, United States of America

Corresponding authors
Vladimir N. Uversky,
vuversky@usf.edu
Elrashdy M.
Redwan, lradwan@kau.edu.sa

## ABSTRACT

There is a continuously increasing pressure associated with the appearance of *Salmonella enterica* Serovar *typhimurium* (*S. typhimurium*) and *Shigella sonnei* (*S. sonnei*) that have developed pathogenic multiple antibiotic resistance and the cost of cure and control of these enterobacteriaceae infections increases annually. The current report for first time demonstrated the distinguished antimicrobial action of camel lactoferrin (cLf) obtained from the milk of different clans of camel in Saudi Arabia against *S. typhimurium* and *S. sonnei*. These cLf subtypes showed comparable antimicrobial potential when tested against the two bacterial strains but were superior to either bovine (bLf) or human lactoferrin (hLf). The synergism between lactoferrins and antibiotics concerning their antibacterial efficacies against the two bacterial strains was evident. Exploring mechanisms by which camel lactoferrin can kill *S. typhimurium* and *S. sonnei* revealed that cLf affects bacterial protein profile. Besides, it interacts with bacterial lipopolysaccharides (LPS) and numerous membrane proteins of *S. typhimurium* and *S. sonnei*, with each bacterial strain possessing distinctive binding membrane proteins for lactoferrin. Furthermore, as evidenced by electron microscopy analysis, cLf induces extracellular and intracellular morphological changes in the test bacterial strains when used alone or in combination treatment with antibiotics. Lactoferrin and antibiotics combination strongly disrupts the integrity of the bacterial cells and their membranes. Therefore, cLf can kill *S. typhimurium* and *S. sonnei* by four different mechanisms, such as iron chelation, affecting some bacterial proteins, binding to bacterial LPS and membrane proteins, and impairing the integrity of the bacterial cells and their membranes.

## INTRODUCTION

*Escherichia coli* strains are not the only cause of foodborne illnesses, such as diarrheal diseases, since various *Shigella* species can annually cause an estimated 165 million cases of severe dysentery (*McCrickard et al., 2018*). Most of these cases occur in young children (under the age of 5 years) living in developing countries, and over one million of the cases reported each year are fatal. *Shigella* species are spread either through direct contact or through polluted water or food intake (*McCrickard et al., 2018*; *Zaidi & Estrada-García, 2014*). Infection with *Shigella* spp. (shigellosis) generates Shiga toxins that invade and damage vascular endothelium and inhibit protein synthesis in host cells in a comparable mode of action to that of ricin; a plant toxin (*Sandvig & van Deurs, 2000*). Also, *Salmonella enterica* can trigger foodborne diseases that have a huge effect on public health (*Havelaar et al., 2015*). About 2,600 closely related serovars characterized by their clustered combination of surface O and H antigens belong to *S. enterica*. There are two *S. enterica* classes based on the various pathogenic habits, namely, typhoidal *S. enterica* and non-typhoidal *S. enterica*. Although typhoidal *Salmonella* may lead to highly fatal systemic infections, non-typhoidal *Salmonella* infections are typically self-limiting (*Gal-Mor, Boyle & Grassl, 2014*). Nevertheless, non-typhoidal *S. enterica* serovar *typhimurium* has developed pathogenic multiple antibiotic resistance, which poses a major threat to public health and food safety. Infection with *S. typhimurium via* contaminated food or water ingestion causes nontyphoidal salmonellosis (gastroenteritis or food poisoning, bacteremia when infection enters bloodstream and succeeding focal infection) (*Majowicz et al., 2010*).

*S. sonnei* developed high resistance to many antibiotics used for the treatment of its infection, such as ampicillin, cephalosporins, gentamicin, piperacillin, tetracycline, ticarcillin, and trimethoprim/sulfamethoxazole (*Wang et al., 2019b*). The emergence of antibiotic resistance patterns in *S. typhimurium* was also reported (*Wang et al., 2019a*). This serovar showed resistance to ampicillin, sulfonamides, streptomycin, tetracycline, and chloramphenicol. Therefore, safe and effective unconventional therapy is strongly recommended for managing these multi-drug resistant and highly transmissible strains of *S. sonnei* and *S. typhimurium*. Many patients are seeking alternative drugs to antibiotics, and Arabian patients mostly turn to camel milk. Camel milk consumption as an essential nutritional and medicinal source is present in common religious beliefs and values of several Arab countries (*Abuelgasim et al., 2018*). In the past two decades, attention from scientists from all over the world has dramatically increased toward researching camel milk and its bioactive ingredients to discover their potential health benefits (*Maghraby, Mohamed & Abdel-Salam, 2005*; *El-Agamy, 2006*; *Shabo, Barzel & Yagil, 2008*; *El-Agamy et al., 2009*; *Alhaj & Kanhal, 2010*; *Khan & Alzohairy, 2011*; *Habib et al., 2013*; *Levy, Steiner & Yagil, 2013*; *Musaad, Faye & Al-Mutairi, 2013*; *Yadav et al., 2015*; *Yassin et al., 2015*; *Jilo & Tegegne, 2016*; *Sumaira Shah et al., 2020*).

Patients with hepatitis C in Egypt use camel milk as both an alternative and a supportive medication (*Redwan & Tabll, 2007*). Therapeutic values of camel milk proteins and what makes them different from proteins of other milk of ruminants have been comprehensively investigated by our research team since 2007. We reported the presence of various bioactive

proteins in camel milk that contribute to its protective and therapeutic potential against hepatitis C virus genotype 4 (*Redwan & Tabll, 2007*; *El-Fakharany et al., 2008*; *Liao et al., 2012*; *El-Fakharany et al., 2013*; *El-Baky, El-Fakharany & Redwana, 2017*), pathogenic bacteria (*Redwan et al., 2016*; *Almehdar et al., 2019*; *Almehdar et al., 2020*; *El-Baky et al., 2021*), and different cancer cells (*Uversky et al., 2017*; *El-Fakharany et al., 2018*; *EL-Baky, Abu-Serie & Redwan, 2021*). These bioactive proteins include antibodies (*EL-Fakharany et al., 2012*), lactoferrin (*Redwan & Tabll, 2007*; *El-Fakharany et al., 2008*; *Liao et al., 2012*; *El-Fakharany et al., 2013*; *El-Baky, El-Fakharany & Redwana, 2017*; *Redwan et al., 2016*; *Almehdar et al., 2019*; *Almehdar et al., 2020*; *EL-Baky, Abu-Serie & Redwan, 2021*), lactoferricin peptide (*El-Baky et al., 2021*), lactoperoxidase (*El-Fakharany, Uversky & Redwan, 2017*), and $\alpha$-lactalbumin (*Uversky et al., 2017*).

We previously demonstrated superior antimicrobial effects of camel lactoferrin on methicillin-resistant *Staphylococcus aureus* (MRSA, a very dangerous form of *Staphylococcus* that becomes resistant to numerous antibiotics) (*Redwan et al., 2016*), pathogenic *E. coli* (*Almehdar et al., 2019*), as well as *Salmonella enterica* serovar Typhi (*S. typhi*) (*Almehdar et al., 2020*) compared to bLf or hLf. We also proved that cLf killed those strains through bacteriostatic and bactericidal mechanisms, either alone or in a synergistic manner with certain antibiotics. The current study aimed to investigate cLf as a possible treatment option against multi-drug resistant and highly transmissible strains of *S. sonnei* and *S. typhimurium* that cause dramatic healthcare problems worldwide and to explore its antimicrobial mechanisms against these pathogens.

## MATERIALS & METHODS

### Lactoferrins

In the present study, hLf (15% iron saturation) and bLf (10% iron saturation) purchased from Sigma-Aldrich (St. Louis, MO, USA) were comparatively evaluated to camel lactoferrins regarding their *in vitro* antimicrobial mechanisms (bacteriostatic and bactericidal activities) against *S. typhimurium* and *S. sonnei*. Protocols of milk collection from four different breeds of Saudi camels, milk processing, cLfs purification, and Lfs labeling with activated NHS-biotin/gold nanoparticles (AuNPs) were reported in earlier studies (*Redwan et al., 2016*; *Almehdar et al., 2019*; *Almehdar et al., 2020*).

### Cultivation conditions

The seed stock cultures of enteric bacterial strains *Salmonella typhimurium* LT2 and *Shigella sonnei* ATCC 25931 were inoculated into tryptic soy broth (TSB). After overnight incubation, a 200 µl inoculum of each enteric bacterial strain was plated on tryptic soy agar (TSA) and incubated overnight. Finally, TSB was inoculated with one of the colonies of each enteric bacterial strain and incubated at 37 °C for 16 h.

### Antimicrobial activity examination using agar disc-diffusion test and monitoring bacterial growth

Standard discs of 17 antibiotics were obtained from Mast Diagnostics (Merseyside, Liverpool, UK) to be used in this assay; amikacin (30 µg), ampicillin (10 µg), augmentin

(30 µg), aztreonam (30 µg), cefoxitin (30 µg), cefepime (30 µg), ceftazidime (30 µg), cephalothin (30 µg), chloramphenicol (30 µg), ciprofloxacin (5 µg), cotrimoxazole (25 µg), fucidic acid (10 µg), gentamicin (10 µg), imipenem (10 µg), oxacillin (1 µg), piperacillin (100 µg), and vancomycin (30 µg).

Antimicrobial activity of four cLfs isolated from milk of four different breeds of Saudi camels (cLf1, cLf2, cLf3, and cLf4), bLf, hLf, and the standard antibiotics against *S. typhimurium* and *S. sonnei* was tested by agar disc-diffusion technique. Plates of Mueller-Hinton (MH) agar were overlaid with $2 \times 10^6$ CFU/ml of *Salmonella typhimurium* LT2 and *Shigella sonnei*. Wells of about 5 mm diameter were made on inoculated plates then different concentrations of lactoferrins (0.00, 0.125, 0.250, 0.50, 0.750, 1, 1.5, 2, 2.5, and 3 mg/ml) were added, and left to diffuse in agar at 4 °C for 2 h before plates finally incubated overnight at 37 °C. The diameters of clear inhibition zones were measured in millimeters. Antibacterial effects of different lactoferrins on growth of *S. typhimurium* and *S. sonnei* after 1, 3, 6, 12, and 24 h of incubation at 37 °C were monitored spectrophotometrically (measuring OD at 620 nm).

## Determination of minimum inhibitory concentrations (MICs) by broth microdilution assay

Cation adjusted Mueller-Hinton (CAMH) broth containing test antibacterial agents in serial dilutions was inoculated with $2 \times 10^6$ CFU/ml of *Salmonella typhimurium* LT2 and *Shigella sonnei* then added to 96-well microtiter plates. Plates were incubated overnight at 37 °C and then absorbance was recorded at 620 nm (monitoring bacterial growth). The lowest concentration of each antibacterial agent at which bacterial growth was completely inhibited was recorded as MIC. Bacteria in CAMH broth was the growth control. To confirm the presence or absence of a ferrochelating-dependent antibacterial mechanism of Lfs, the MICs for all Lfs were evaluated once more after iron supplementation (adding ferrinitrilotriacetate (FeNTA) at a concentration of 50 µM) at the time of inoculation (*Redwan et al., 2016*; *Almehdar et al., 2019*; *Almehdar et al., 2020*).

## Checkerboard assay

To study synergistic interaction between the different lactoferrins and antibiotics regarding their antibacterial activity against *Salmonella typhimurium* LT2 and *Shigella sonnei*, combinations of cLf-chloramphenicol, cLf-cefepime, cLf-imipenem, hLf-chloramphenicol, hLf-cefepime, hLf-imipenem, bLf-chloramphenicol, bLf-cefepime, and bLf-imipenem were examined and compared with their individual antibacterial activities *via* microdilution checkerboard method as described previously (*Redwan et al., 2016*; *Almehdar et al., 2019*; *Almehdar et al., 2020*). The combined antibacterial effects of cLf, hLf, and bLf with each studied antibiotic were analyzed by calculation of fractional inhibitory concentration index (FICI) (*Redwan et al., 2016*; *Almehdar et al., 2019*; *Almehdar et al., 2020*). Firstly, we calculated the fractional inhibitory concentration (FIC) of each lactoferrin or antibiotic by dividing MIC value of the agent in combination by MIC value of the agent alone. Afterward, FIC value of each lactoferrin was divided by FIC of the antibiotic to estimate FICI. The FICI values of 0.5 indicate that the MIC values of both antibiotic and Lf in
the combination treatment were reduced to $\frac{1}{4}$ MIC. The FICI values of 0.37 indicate that MICs of antibiotics were reduced to $\frac{1}{4}$ MIC and MICs of Lfs were reduced to 1/8 MIC. The antibacterial combination against *S. typhimurium* and *S. sonnei* was considered indifferent at $0.5 <$ FICI $< 4$, antagonistic at FICI $\geq 4$, and synergistic at FICI $\leq 0.5$.

### Time-kill assay

Assays for the killing rate of *S. typhimurium* and *S. sonnei* by the different lactoferrins either alone or combined with antibiotics were performed as previously reported (*Redwan et al., 2016*; *Almehdar et al., 2019*; *Almehdar et al., 2020*). Each lactoferrin at concentrations of $\frac{1}{4}$ MIC or $1\times$ MIC was combined with chloramphenicol, imipenem, or cefepime at concentration of $\frac{1}{4}$ MIC. Lactoferrins alone were added at concentrations of $\frac{1}{2}\times$, $1\times$, and $2 \times$ MIC in LB broth. Antimicrobials were added to *S. typhimurium* and *S. sonnei* ($2\times10^6$ CFU/ml) cultures followed by incubation at 37 °C and 150 rpm. Viable colony counts were done at 0, 4, and 8 h. Experiments were conducted three times and the obtained results were demonstrated as mean and the standard error of the mean (SEM). Bactericidal activity was defined as $\geq 3 \log_{10}$ CFU/ml reduction (99.9% kill) from the count of the control culture (without addition of any antimicrobial).

### Calculation of minimum bactericidal concentration

All MIC wells from broth microdilution assay with no bacterial growth were subcultured on plates of MH agar that did not contain test lactoferrins and incubated at 37 °C for 24 h. Bacterial viability was measured by the count of colonies. Bacterial growth resumption means that test lactoferrins did not cause bacterial death, thus identified as bacteriostatic, while absence of bacterial proliferation means that test lactoferrins caused bacterial death, thus identified as bactericidal. The lowest concentration of test lactoferrins resulting in $a \geq 99.9\%$ decrease in viable cells (bacterial death) was recorded as minimum bactericidal concentration (MBC). Generally, antibacterial compounds are considered bactericidal if their MBC value is at most four times their MIC value (*French, 2006*).

### Analysis of interaction between the biotinylated lactoferrins and *S. typhimurium/S. sonnei* LPS

Reactivity of Lfs with LPS of *S. typhimurium* or *S. sonnei* obtained from Sigma Aldrich (St. Louis, MO, USA) was estimated using ELISA as previously described (*Almehdar et al., 2019*; *Almehdar et al., 2020*).

### Effects of the different antibacterial agents (antibiotics, cLf, and their combination) on the integrity of *S. typhimurium/S. sonnei* cell and its protein profile

Cell pellets of 5 ml *S. typhimurium* and *S. sonnei* cultures were collected then suspended in 5 ml phosphate buffered saline (PBS) buffer, pH 7.4. Suspended bacterial cells were treated with either carbenicillin, chloramphenicol, or imipenem alone, or combined with cLf, or cLf alone, and incubated at 37 °C. Samples (200 µl) were withdrawn from each treatment at different time intervals (0, 30, 60, and 150 min). These samples were split into two halves, the first one was centrifuged and its corresponding supernatant protein content

was evaluated by Bradford assay kit at 595 nm (Thermo Fisher Scientific, Waltham, MA, USA). The second half was used for SDS-PAGE protein profiling (*Almehdar et al., 2019*; *Almehdar et al., 2020*).

### Separation of bacterial membrane proteins fraction and analysis of their interactions with cLf-biotin

*S. typhimurium* and *S. sonnei* cells were pelleted and then washed with PBS, pH 7.4. The membrane proteins fraction of bacterial cells were then separated and run on 12.5% reducing SDS-PAGE as described previously (*Redwan et al., 2016*; *Almehdar et al., 2019*; *Almehdar et al., 2020*). The resolved bacterial membrane proteins (BMP) of *S. typhimurium* or *S. sonnei* on SDS-PAGE were transferred into poly-vinylidene fluoride (PVDF) membrane by western blotting then incubated with cLf-biotin. Afterwards, the membrane was incubated with streptavidin-peroxidase and developed with peroxidase substrate. The interaction between separated bacterial membrane proteins and cLf was analyzed by ELISA as described in previous studies (*Redwan et al., 2016*; *Almehdar et al., 2019*; *Almehdar et al., 2020*). After coating of 96-well ELISA microtiter plate with the blank buffer of carbonate/bicarbonate pH 9.6, free camel lactoferrin and BMP extract of *S. typhimurium*/*S. sonnei* were added. Subsequently, the plate was incubated with biotinylated cLf then streptavidin-peroxidase, and developed with TMP; peroxidase substrate. The plate OD was recorded at 450 nm, the results were demonstrated as mean $\pm$ SD of eight replicates.

### Scanning electron microscopy (SEM) and transmission electron microscopy (TEM)

To track morphological changes both extracellularly and intracellularly after *S. typhimurium* and *S. sonnei* treatment by an antibiotic (carbenicillin) or cLf alone (labeled with AuNPs) and in combination, scanning and transmission electron microscopy were used as previously reported (*Almehdar et al., 2019*; *Almehdar et al., 2020*).

### Protein content evaluation

The content of test proteins (lactoferrins either labeled or free) was estimated by Bradford assay. Determination of the remaining proteins present in the supernatant solution from combining washes was used to indirectly evaluate the labeled proteins. The amount of bound or labeled protein was determined by difference spectrophotometry.

### Statistical analysis

The experiments were carried out in triplicate and the obtained results were demonstrated as mean $\pm$ SD of triplicate. Data analysis was done by using Student's $t$-test and McNemar's test. A $P$-value of less than 0.01 was regarded as statistically significant.

## RESULTS

### Antimicrobial activity examination

*Salmonella typhimurium* LT2 was susceptible to 13 of the 17 tested antibiotics, but showed resistance to vancomycin, fucidic acid, oxacillin, and ampicillin (Table 1). While *S. sonnei*

**Table 1** Testing susceptibility of *S. typhimurium* or *S. sonnei* to a panel of 17 antibiotics by the agar disc-diffusion technique.

| Antibacterial agent | Concentration of antibacterial agent (µg/disc) | Average diameter of inhibition zone ( ± 1 mm) | |
|---|---|---|---|
| | | *S. typhimurium* | *S. sonnei* |
| Gentamicin (GM) | 10 | 24 | 20 |
| Vancomycin (VA) | 30 | R | 15 |
| Fucidic acid (FC) | 10 | R | 10 |
| Chloramphenicol (C) | 30 | 25 | 30 |
| Oxacillin (OX) | 1 | R | R |
| Cefepime (CPM) | 30 | 35 | 25 |
| Ampicillin (AP) | 10 | R | R |
| Augmentin (AUG) | 30 | 15 | R |
| Cefoxitin (FOX) | 30 | 16 | R |
| Cephalothin (KF) | 30 | 12 | 25 |
| Cotrimoxazole (TS) | 25 | 26 | 26 |
| Amikacin (AK) | 30 | 30 | 20 |
| Ceftazidime (CAZ) | 30 | 30 | R |
| Aztreonam (ATM) | 30 | 33 | R |
| Piperacillin (PRL) | 100 | 33 | R |
| Imipenem (IMI) | 10 | 33 | 12 |
| Ciprofloxacin (CIP) | 5 | 33 | 17 |

**Notes.**
Abbreviations: R, Resistant (no inhibition zone).

ATCC 25931 was susceptible to 10 of the 17 tested antibiotics and showed resistance to oxacillin, ampicillin, augmentin, cefoxitin, ceftazidime, aztreonam, and piperacillin (Table 1). The corresponding experiments showing the microbial resistance to antibiotics were conducted to confirm that we are carrying out our study on the pathogenic multiple antibiotic resistant strains of *S. typhimurium* and *S. sonnei*. The *in vitro* antibacterial activity of cLf subtypes isolated from different clans of Saudi camels (cLf1, cLf2, cLf3, and cLf4), bLf, or hLf against *S. typhimurium* and *S. sonnei* was evaluated by measuring their inhibition zones at different Lf concentrations in agar disc-diffusion assay. All lactoferrins efficiently inhibited *S. typhimurium* and *S. sonnei* growth, producing concentration-dependent inhibition zones (Table 2). We noticed that bLf and hLf were able to inhibit *S. typhimurium* growth at concentrations ranging from 0.75 to 3 mg/ml and *S. sonnei* growth at concentrations ranging from 1 to 3 mg/ml, whereas all camel lactoferrins could inhibit the growth of both bacterial strains at concentrations ranging from 0.125 to 3 mg/ml (Table 2). This suggests the superiority (six and eight folds) of cLf antibacterial activity against *S. typhimurium* and *S. sonnei*, respectively over that of both hLf and bLf. No significant differences in antimicrobial activity were observed between cLf subtypes.

## Inhibitory effects of lactoferrins on *S. typhimurium* and *S. sonnei*

After incubation of *S. typhimurium* and *S. sonnei* with various concentrations of bLf, hLf, and different cLfs, it was observed that all cLfs inhibited the growth of both bacterial
**Table 2** Testing susceptibility of *S. typhimurium* and *S. sonnei* to different lactoferrins at different concentrations.

| Lf | Concentration (mg/ml) | Average diameter of inhibition zone (±1 mm) | |
|---|---|---|---|
| | | *S. typhimurium* | *S. sonnei* |
| bLf | 3 | 53 | 50 |
| | 2.5 | 48 | 47 |
| | 2 | 40 | 36 |
| | 1.5 | 33 | 27 |
| | 1 | 24 | 17 |
| | 0.75 | 10 | R |
| | 0.5 | R | R |
| | 0.25 | R | R |
| | 0.125 | R | R |
| hLf | 3 | 52 | 48 |
| | 2.5 | 46 | 42 |
| | 2 | 41 | 37 |
| | 1.5 | 38 | 29 |
| | 1 | 27 | 20 |
| | 0.75 | 12 | R |
| | 0.5 | R | R |
| | 0.25 | R | R |
| | 0.125 | R | R |
| cLf1 | 3 | 55 | 51 |
| | 2.5 | 50 | 48 |
| | 2 | 48 | 44 |
| | 1.5 | 43 | 39 |
| | 1 | 39 | 34 |
| | 0.75 | 35 | 30 |
| | 0.5 | 30 | 27 |
| | 0.25 | 26 | 21 |
| | 0.125 | 18 | 15 |
| cLf2 | 3 | 57 | 52 |
| | 2.5 | 50 | 47 |
| | 2 | 45 | 44 |
| | 1.5 | 40 | 40 |
| | 1 | 36 | 35 |
| | 0.75 | 31 | 30 |
| | 0.5 | 28 | 27 |
| | 0.25 | 23 | 24 |
| | 0.125 | 19 | 16 |

**Table 2** (*continued*)

| Lf | Concentration (mg/ml) | Average diameter of inhibition zone (±1 mm) | |
| --- | --- | --- | --- |
| | | *S. typhimurium* | *S. sonnei* |
| cLf3 | 3 | 55 | 50 |
| | 2.5 | 50 | 47 |
| | 2 | 44 | 43 |
| | 1.5 | 40 | 38 |
| | 1 | 36 | 32 |
| | 0.75 | 30 | 28 |
| | 0.5 | 25 | 24 |
| | 0.25 | 23 | 20 |
| | 0.125 | 21 | 17 |
| cLf4 | 3 | 56 | 53 |
| | 2.5 | 52 | 50 |
| | 2 | 48 | 47 |
| | 1.5 | 42 | 40 |
| | 1 | 39 | 36 |
| | 0.75 | 34 | 31 |
| | 0.5 | 28 | 27 |
| | 0.25 | 23 | 22 |
| | 0.125 | 18 | 18 |

**Notes.**

Abbreviations: R, Resistant (no inhibition zone).

strains completely after incubation for 3–24 h at concentrations of 1–3 mg/ml, 6–24 h at a concentration of 0.75 mg/ml, 12–24 h at a concentration of 0.5 mg/ml, and 24 h at concentrations of 0.125−0.25 mg/ml (Table S1). On the other hand, bLf and hLf could inhibit *S. typhimurium* growth completely after incubation for 3–24 h at concentrations of 1.5–3 mg/ml, 6–24 h at a concentration of 1 mg/ml, and 12–24 h at a concentration of 0.75 mg/ml. Both bLf and hLf also inhibited *S. sonnei* growth completely after incubation for 3–24 h at concentrations of 2–3 mg/ml, 6–24 h at a concentration of 1.5 mg/ml, and 12–24 h at a concentration of 1 mg/ml. No antibacterial effects were noticed after incubation with hLf or bLf at concentrations of 0.125−0.5 mg/ml in the case of *S. typhimurium* and at concentrations of 0.125−0.75 mg/ml in case of *S. sonnei*, suggesting superior activity of camel lactoferrins (Table S1). Furthermore, the data presented in Table S1 demonstrate that the culture OD decreases over time in the presence of lactoferrins before reaching complete inhibition of growth suggesting that Lfs not only inhibit bacterial growth but also induce cell death *via* lysis. Since no difference was found between the different types of camel lactoferrin in their inhibitory effectiveness, we used only cLf1 in the remaining parts of this work.

## Determination of MIC values of antibacterial agents by broth microdilution assay

The MIC values of cLf, and hLf or bLf against *S. typhimurium* were calculated to be 125 and 750 µg/ml, respectively. While MIC values of cLf, and hLf or bLf against *S. sonnei*

were calculated to be 0.125 and 1 mg/ml, respectively. These values confirm the superior (six and eight folds against *S. typhimurium* and *S. sonnei*, respectively) antibacterial effect of cLf. MIC values determined for chloramphenicol, cefepime, and imipenem against *S. typhimurium* and *S. sonnei* were 1.3, 1.7, and 1.9 μg/ml, respectively. We also found that the inhibitory activity of lactoferrins was reversed after addition of iron as FeNTA to bacterial cultures. The MIC values obtained for cLf and hLf or bLf against *S. typhimurium* after iron addition were 1 and 3 mg/ml, respectively, and against *S. sonnei* were 1 and 4 mg/ml, respectively. The reported here reverse of the inhibitory activity means that Lf antimicrobial mechanisms include exhibiting bacteriostatic effect based on the Lf ability to bind free iron (at concentration lower than $2 \times$ MIC), and this bacteriostatic effect can be reversed by iron supplementation as observed in this study and our previous work (*Redwan et al., 2016*; *Almehdar et al., 2019*; *Almehdar et al., 2020*).

## Checkerboard assay

The synergy between the studied antibiotics and cLf, hLf, or bLf against *S. typhimurium* and *S. sonnei* was evident from the corresponding FICI values ranging from 0.37 to 0.5 (Table 3). This was further explored by examining the changes induced in the morphology of bacterial cells caused by these agents alone or combined under electron microscopy.

## Time-kill assay

Figure 1 demonstrates the time-kill study data as changes in the $\log_{10}$ CFU/ml of *S. typhimurium* and *S. sonnei* after either 4 h or 8 h of incubation with each Lf alone, and combined with different antibiotics. The reduction in viable cell count was greatest after 8 h of incubation with cLf combined with chloramphenicol ($-4.334 \log_{10}$ and $-4.230 \log_{10}$) in the case of *S. typhimurium* and *S. sonnei*, respectively. Also, it was observed that all combinations were significantly more effective at reducing viable bacteria count than Lf alone even at a concentration of $2\times$ MIC (Fig. 1).

## Calculation of minimum bactericidal concentration

All tested Lfs can be considered as bacteriostatic agents against *S. typhimurium* and *S. sonnei* after resuming bacterial growth of MIC wells with no growing bacteria. However, cLf and hLf/bLf achieved bactericidal activity against *S. typhimurium* and *S. sonnei* only at a concentration of $2 \times$ MIC; 0.25 mg/ml and 1.5 mg/ml, respectively in the case of *S. typhimurium* and 0.25 mg/ml and 2 mg/ml, respectively in case of *S. sonnei*. Since the ratio of MBC/MIC for all test lactoferrins was equal to two, they were labeled as bactericidal agents. These data also confirm the superior antibacterial activity of cLf.

## Analysis of interaction between the biotinylated lactoferrins and *S. typhimurium*/*S. sonnei* LPS

We failed many times in running LPS of *S. typhimurium* or *S. sonnei* on SDS-PAGE then transferring them to nitrocellulose blots, and only ELISA could detect LPS possibly because of the interference of SDS with the bacterial LPS as we mentioned in previous studies (*Almehdar et al., 2019*; *Almehdar et al., 2020*). Nevertheless, silver nitrate could stain LPS of *S. typhimurium* and *S. sonnei* on SDS-PAGE (data not shown).

**Table 3    Effect of different combinations of lactoferrins and antibiotics against *S. typhimurium* and *S. sonnei*.**

| Antimicrobial combination | FICI | | Effect | |
|---|---|---|---|---|
| | *S. typhimurium* | *S. sonnei* | *S. typhimurium* | *S. sonnei* |
| cLf-C | 0.37 | 0.37 | Synergy | Synergy |
| hLf-C | 0.5 | 0.37 | Synergy | Synergy |
| bLf-C | 0.5 | 0.37 | Synergy | Synergy |
| cLf-CPM | 0.37 | 0.37 | Synergy | Synergy |
| hLf-CPM | 0.5 | 0.37 | Synergy | Synergy |
| bLf-CPM | 0.37 | 0.37 | Synergy | Synergy |
| cLf-IMI | 0.37 | 0.5 | Synergy | Synergy |
| hLf-IMI | 0.5 | 0.5 | Synergy | Synergy |
| bLf-IMI | 0.37 | 0.5 | Synergy | Synergy |

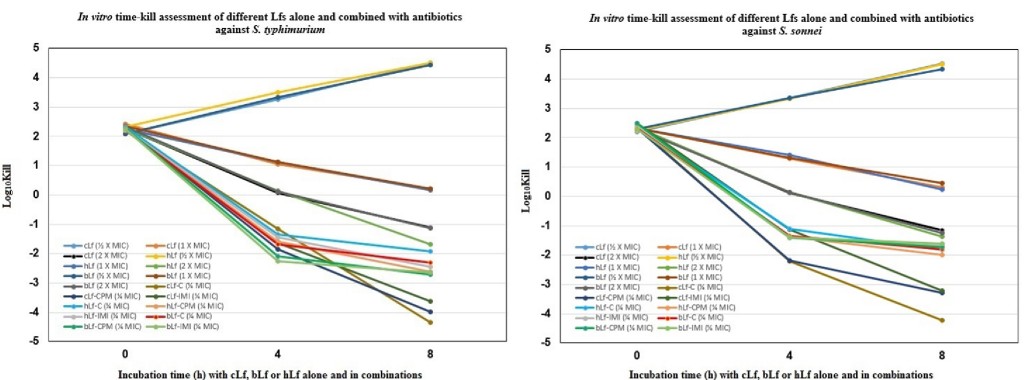

**Figure 1    *In vitro* time-kill assessment of different Lactoferrins alone and combined with antibiotics against *S. typhimurium* and *S. sonnei*.**

ELISA results of lactoferrins interaction with *S. typhimurium* or *S. sonnei* LPS demonstrated a *significant* ($p < 0.01$) *interaction* between all tested Lfs and *S. typhimurium* or *S. sonnei* LPS (Fig. 2). Camel lactoferrin was more reactive ($p < 0.001$) than bLf and hLf against *S. typhimurium* or *S. sonnei* LPS, with no evident differences between the cLf sub-types. These observations reveal that the cLf sub-types (cLf1, cLf2, cLf3, and cLf4) isolated from milk of four different breeds of Saudi camels have possibly no differences in their glycosylation moieties that are involved in the interaction between the LPS and lactoferrin, which needs further comprehensive investigation.

## Effect of the different antibacterial agents on the integrity of the *S. typhimurium*/*S. sonnei* cells and their protein profile

Table 4 and Fig. 3 reveal the effect of antibiotics (carbenicillin, chloramphenicol, or imipenem), cLf, and cLf-antibiotics combinations on the integrity of *S. typhimurium* or *S. sonnei* cell (protein released from treated cells was measured by Bradford assay at 595 nm). Furthermore, Table 5 described the effect of antibiotics, cLf, and their combinations

**Figure 2  Analysis of interaction between the biotinylated lactoferrins and *S. typhimurium*/*S. sonnei* LPS using ELISA.** After coating of ELISA plate with either *S. typhimurium* or *S. sonnei* LPS, biotinylated lactoferrins were added. Following washing to remove unbound biotinylated lactoferrins, streptavidin-peroxidase was added. The plate was washed again, then incubated with peroxidase substrate (TMP). The plate OD was recorded at 490 nm. Results demonstrated a significant ($p < 0.01$) interaction between all tested Lfs and *S. typhimurium* or *S. sonnei* LPS compared to blank (carbonate/bicarbonate pH 9.6 coating buffer). Camel lactoferrin was more reactive ($p < 0.001$) than bLf and hLf against *S. typhimurium* or *S. sonnei* LPS.

on protein profile of *S. typhimurium* or *S. sonnei* cell (protein appearance, disappearance, and/or high expression). Protein profile of *S. typhimurium* or *S. sonnei* cell clearly changed on SDS-PAGE over time (from 30 min to 150 min) depending on the used antibacterial agents in treatment. These results obviously indicate that cLf and antibiotics either alone or in combination exert significant ($p < 0.01$, and $p < 0.001$) effect on *S. typhimurium* or *S. sonnei* cell integrity (impairing the integrity of the bacterial cells and their membranes, which lead to release of cell protein content) and change bacterial protein profile. Additionally, these effects increase over time. The presence of synergism between cLf and antibiotics with regard to their effect on the cell integrity and protein profile of the two bacterial strains was also observed.

## Separation of bacterial membrane proteins fraction and analysis of their interactions with cLf-biotin

The BMP extract of *S. typhimurium* or *S. sonnei* was resolved on SDS-PAGE (Fig. 4). The profile of membrane proteins was found to be different from the profile of whole-cell lysate (Fig. 4). Results of western blotting confirmed the interaction of cLf-biotin with multiple

**Table 4** Protein content released from *S. typhimurium* or *S. sonnei* treated by various antibacterials (antibiotics, cLf, and their combination) at different time intervals and measured by Bradford assay at 595 nm.

| | Effect of carbenicillin (CB) on *S. typhimurium* or *S. sonnei* integrity | | | | | | | |
|---|---|---|---|---|---|---|---|---|
| **Antibacterial agents** | ***S. typhimurium*** | | | | ***S. sonnei*** | | | |
| | Time of incubation | | | | Time of incubation | | | |
| | **0 min** | **30 min** | **60 min** | **150 min** | **0 min** | **30 min** | **60 min** | **150 min** |
| CB | $0.22 \pm 0.1$ | $0.56 \pm 0.04$ | $0.75 \pm 0.08$ | $0.86 \pm 0.14$ | $0.2 \pm 0.05$ | $0.35 \pm 0.03$ | $0.44 \pm 0.11$ | $0.57 \pm 0.13$ |
| cLf | $0.32 \pm 0.08$ | $0.61 \pm 0.09^{*}$ | $0.83 \pm 0.09^{*}$ | $1.11 \pm 0.09^{*}$ | $0.35 \pm 0.05$ | $0.61 \pm 0.07^{*}$ | $0.85 \pm 0.07^{*}$ | $0.99 \pm 0.12^{*}$ |
| CB-cLf | $0.35 \pm 0.05$ | $0.75 \pm 0.06^{**}$ | $0.94 \pm 0.11^{**}$ | $1.5 \pm 0.1^{**}$ | $0.42 \pm 0.08$ | $0.69 \pm 0.1^{**}$ | $0.97 \pm 0.08^{**}$ | $1.0 \pm 0.1^{**}$ |
| | Effect of chloramphenicol (C) on *S. typhimurium* or *S. sonnei* integrity | | | | | | | |
| C | $0.19 \pm 0.02$ | $0.33 \pm 0.06$ | $0.55 \pm 0.07$ | $0.78 \pm 0.12$ | $0.21 \pm 0.08$ | $0.57 \pm 0.11$ | $0.68 \pm 0.07$ | $0.95 \pm 0.14$ |
| cLf | $0.3 \pm 0.03$ | $0.41 \pm 0.09^{*}$ | $0.65 \pm 0.05^{*}$ | $0.81 \pm 0.19^{*}$ | $0.21 \pm 0.04$ | $0.72 \pm 0.17^{*}$ | $0.8 \pm 0.18^{*}$ | $1.51 \pm 0.17^{*}$ |
| C-cLf | $0.37 \pm 0.04$ | $0.62 \pm 0.08^{**}$ | $0.78 \pm 0.12^{**}$ | $1.4 \pm 0.21^{**}$ | $0.3 \pm 0.02$ | $0.74 \pm 0.15^{**}$ | $0.91 \pm 0.09^{**}$ | $2.1 \pm 0.28^{**}$ |
| | Effect of Imipenem (IMI) on *S. typhimurium* or *S. sonnei* integrity | | | | | | | |
| IMI | $0.22 \pm 0.08$ | $0.75 \pm 0.15$ | $0.8 \pm 0.2$ | $1.1 \pm 0.17$ | $0.2 \pm 0.05$ | $0.48 \pm 0.04$ | $0.87 \pm 0.11$ | $0.9 \pm 0.1$ |
| cLf | $0.36 \pm 0.04$ | $0.91 \pm 0.09^{*}$ | $1.3 \pm 0.22^{*}$ | $1.9 \pm 0.4^{*}$ | $0.34 \pm 0.06$ | $0.7 \pm 0.17^{*}$ | $0.9 \pm 0.19^{*}$ | $0.93 \pm 0.16^{*}$ |
| IMI-cLf | $0.3 \pm 0.08$ | $1.4 \pm 0.22^{**}$ | $1.91 \pm 0.34^{**}$ | $2.7 \pm 0.4^{**}$ | $0.37 \pm 0.11$ | $0.88 \pm 0.17^{**}$ | $0.99 \pm 0.13^{**}$ | $1.1 \pm 0.25^{**}$ |

Notes.

Data are represented as mean $\pm$ SD of three replicates.

*pointed the significance of values, $p < 0.01$.

**pointed the significance of values, $p < 0.001$.

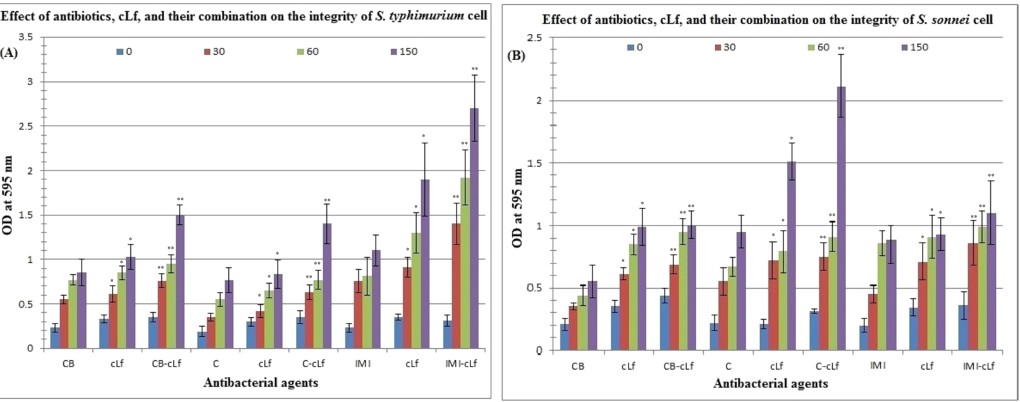

**Figure 3** Effect of antibiotics, cLf, and their combination on the integrity of *S. typhimurium* (A) or *S. sonnei* (B) cell. Protein content released from bacterial cells treated by various antibacterials at different time intervals (0, 30, 60, and 150 min) was measured by Bradford assay at 595 nm. * and ** indicate the significance of values, $p < 0.01$, and $p < 0.001$, respectively.

protein receptors of *S. typhimurium* or *S. sonnei*, and Table 6 summarizes the bacterial membrane proteins of *S. typhimurium* or *S. sonnei* that could interact with cLf, and showed that each strain has its distinctive binding membrane proteins for cLf.

ELISA plate coated with BMP crude extract of *S. typhimurium* or *S. sonnei* exposed to cLf-biotin showed significant signals ($p < 0.001$) (Fig. 5). From *these data* and those of Lf-LPS interaction analysis (Fig. 2), we propose that cLf could kill *S. typhimurium* and *S.*

**Table 5** Molecular weights of changed (appeared, disappeared, and/or highly expressed) bacterial proteins in profile of *S. typhimurium* or *S. sonnei* treated by different test antibacterials on SDS-PAGE taken at different time intervals.

| | Molecular weights (kDa) of *S. typhimurium* or *S. sonnei* proteins changed by carbenicillin (CB) | | | | | | | |
|---|---|---|---|---|---|---|---|---|
| Antibacterial agents | *S. typhimurium* | | | | *S. sonnei* | | | |
| | Time of incubation | | | | Time of incubation | | | |
| | 0 min | 30 min | 60 min | 150 min | 0 min | 30 min | 60 min | 150 min |
| CB | 0 | 62, 57, 38 | 62,57,38 | >175, 62,57,38 | 0 | 72,62,50 | 72,62,50 | 72,62,50 |
| cLf | 0 | 80, 33 | 80, 33 | 80, 33 | 0 | 95,62 | 95,62 | 95,62,51,47 |
| CB-cLf | 0 | 90, 51, 22 | 90, 51, 22 | 90, 51, 22 | 0 | 62, 55, 42 | 62, 55, 42, 25,22, 17, 14 | 62, 55, 42, 25,22, 17, 14 |
| | Molecular weights (kDa) of *S. typhimurium* or *S. sonnei* proteins changed by chloramphenicol (C) | | | | | | | |
| C | 0 | 56, 48, 35 | 56, 48, 35 | 48, 35 | 0 | 30 | 30 | 30 |
| C-cLf | 0 | 62, 22 | 62, 22 | 62, 22 | 0 | 42, 30 | 42, 30 | 42, 30 |
| | Molecular weights (kDa) of *S. typhimurium* or *S. sonnei* proteins changed by Imipenem (IMI) | | | | | | | |
| IMI | 0 | 45, 42, 19 | 45, 42, 19 | 45, 42, 19 | 0 | 51, 44, 19 | 51, 44, 19 | 51, 44, 19 |
| IMI-cLf | 0 | 22, 14, 12 | 22, 14, 12 | 22, 14, 12 | 0 | 62, 52, 39, 22 | 62, 52, 39, 22 | 62, 52, 39, 22 |

**Notes.**
All stained gels were analyzed manually.

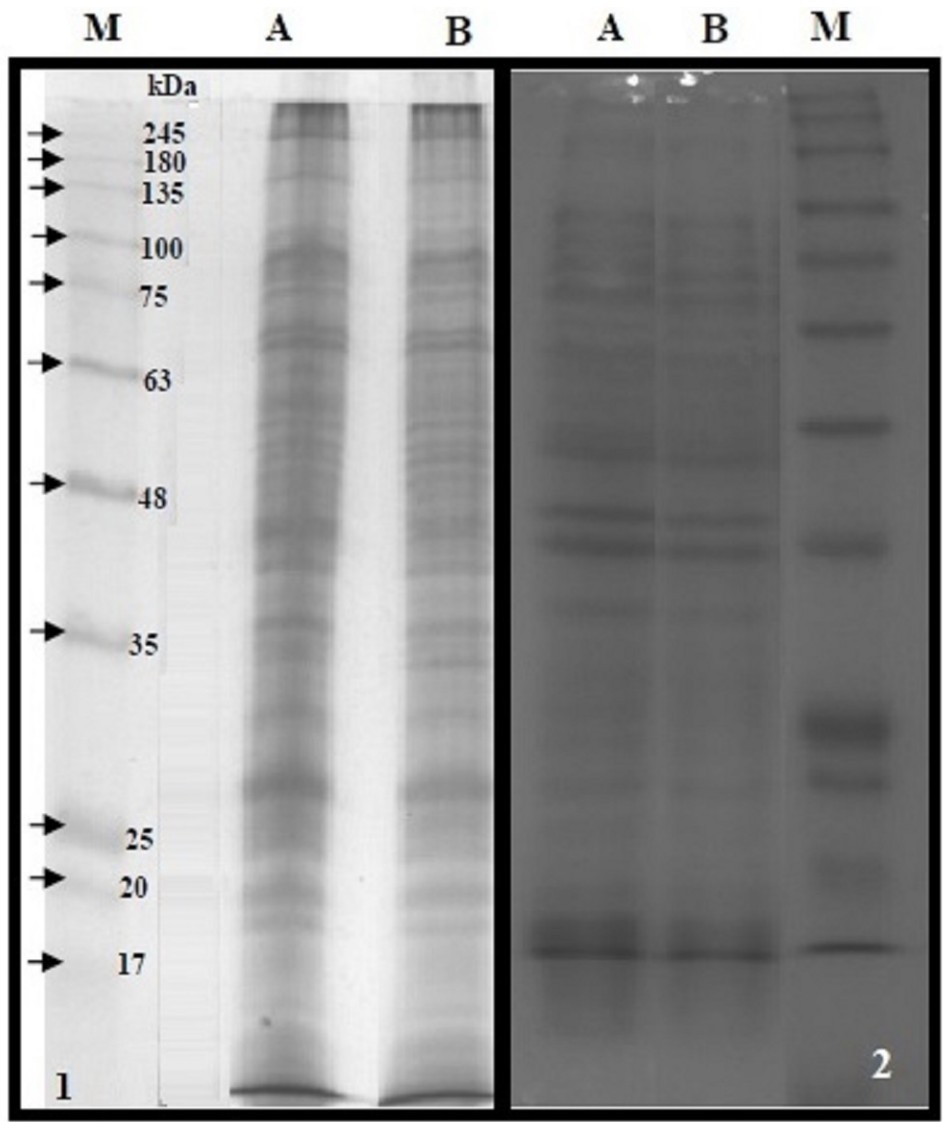

**Figure 4** **SDS-PAGE of the BMP extract and proteins of whole cell lysate.** 12.5% reducing SDS-PAGE of the BMP extract of *S. typhimurium* (panel 1A) or *S. sonnei* (panel 1B) and proteins of whole cell lysate of *S. typhimurium* (panel 2A) or *S. sonnei* (panel 2B). The M lanes in both panels represent the protein molecular weight marker.

*sonnei via* a bactericidal mechanism that involves binding to bacterial LPS and membrane proteins.

## SEM and TEM analyses

The SEM examination of *S. typhimurium* in Figs. 6A–6D and *S. sonnei* in Figs. 7A–7D demonstrated that treated bacterial cells showed irregular and wrinkled outer surfaces, adhesion and aggregation of damaged cells, fragmentation, or the presence of crushed cells and cellular debris. These observations were more evident with cLf-carbenicillin treated cells indicating a clear synergistic effect of both agents on *S. typhimurium* and *S. sonnei*. On

**Table 6  Recognized membrane proteins of bacterial strains by cLf.**

| | Molecular weights (kDa) on SDS-PAGE/trans-blot[a] of binding membrane proteins for cLf |
|---|---|
| **Bacterial strains** | **Membrane proteins** |
| *S. typhimurium* | 180, 64 |
| *S. sonnei* | >245, 30–35, 60–70 |

**Notes.**
[a]The trans-blot was repeated three times.

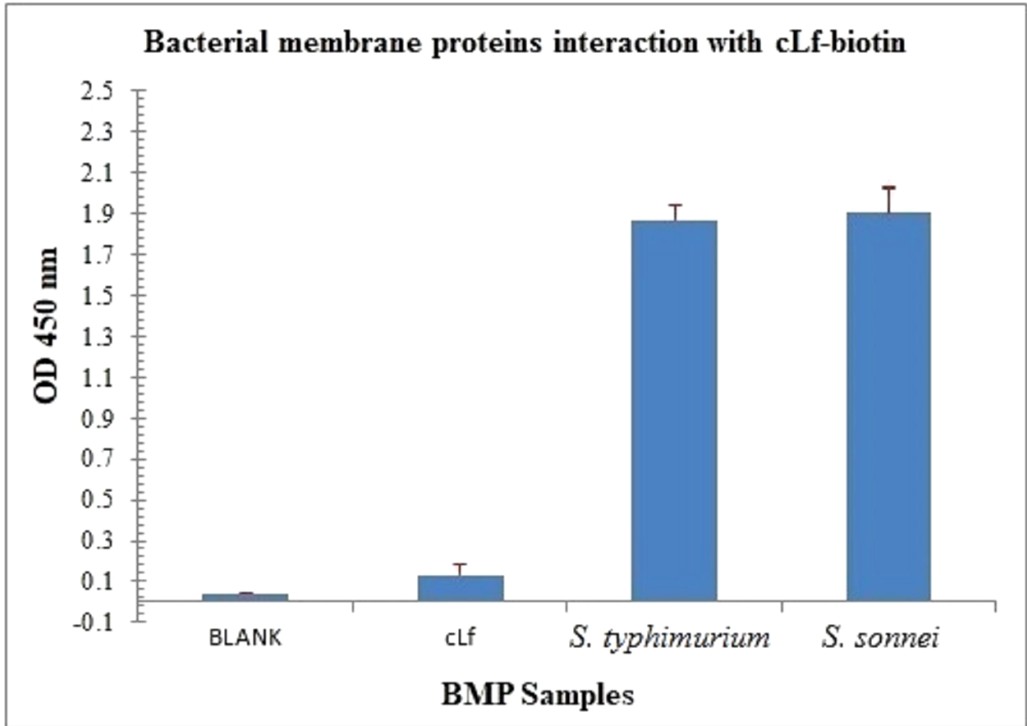

**Figure 5  Immunoassay estimation of interaction between cLf-biotin and BMP extract of *S. typhimurium* or *S. sonnei*.** The plate OD was recorded at 450 nm, the results were demonstrated as mean ± SD of eight replicates. ELISA plate coated with BMP crude extract of *S. typhimurium* or *S. sonnei* exposed to cLf-biotin showed significant signals ($p < 0.001$) compared to blank buffer of carbonate/bicarbonate pH 9.6 or free camel lactoferrin.

the other hand, TEM experiments (Figs. 6E–6H and Figs. 7E–7H) showed deformation of cell membrane and wall, and the gold nanoparticles (black dots) can be easily observed both on and inside the treated cells. The cLf-carbenicillin combination treatment showed the clearest deformation signs; disintegration of cell membrane and wall, bigger cell size, loss of regular cellular shape, and separation of the cell wall (manifested as zigzag or erosion) from the cell.

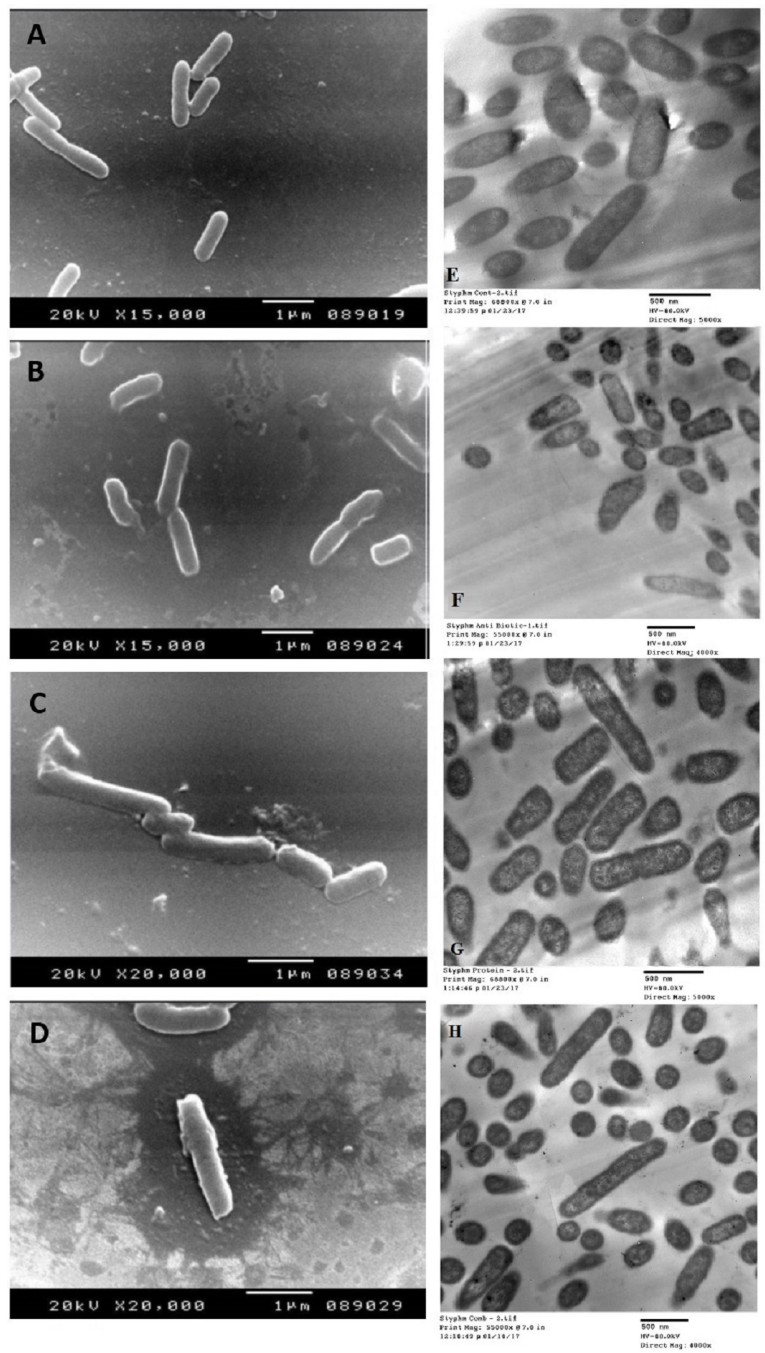

**Figure 6** **Scanning electron microscope images (A–D) and transmission electron microscope images (E–H).** (A) Untreated *S. typhimurium*; (B) Antibiotic (carbenicillin)-treated *S. typhimurium*; (C) cLf-treated *S. typhimurium*; (D) Synergistic effects of cLf-carbenicillin treatment on *S. typhimurium*.; (E) Untreated *S. typhimurium*; (F) Antibiotic (carbenicillin)-treated *S. typhimurium*; (G) cLf-treated *S. typhimurium*. Black dots (gold nanoparticles) scattered extracellularly and in intracellular compartments besides the destructive effects of Lf; (H) Combined effect of cLf-carbenicillin on *S. typhimurium*. Black gold particles are scattered extracellularly and in intracellular compartments in addition to the vigorous destructive and membrane fragmentation effects.

## DISCUSSION

In 2007, we started a systematic research on camel milk and its protective antimicrobial proteins and obtained significant and valuable outcomes especially regarding camel lactoferrin (*Redwan & Tabll, 2007*; *El-Fakharany et al., 2008*; *EL-Fakharany et al., 2012*; *El-Fakharany et al., 2013*; *El-Baky, El-Fakharany & Redwana, 2017*; *Redwan et al., 2016*; *Almehdar et al., 2019*; *Almehdar et al., 2020*; *EL-Baky, Abu-Serie & Redwan, 2021*; *Redwan, 2009*; *Ng et al., 2011*; *Redwan et al., 2014*). We started this line of research mainly because of the common use of camel milk and even camel urine as an alternative medicine for hepatitis C treatment in the Egyptian community. So, the target was to find the scientific basis for the use of this alternative medicine. The *in vitro* data confirmed the significant antiviral activity of camel lactoferrin (in its native and recombinant full-length forms or even its N and C lobes) against hepatitis C virus (*Liao et al., 2012*). We even performed a clinical study in one of Egypt Governorates involving camel milk as a treatment for patients with hepatitis C after stopping any other treatment for 6 months and obtained results that were encouraging (*El-Fakharany et al., 2017*).

Then we moved to the exploration of antibacterial activities of cLf isolated from milk of camels in one of Egypt farms and compared its activites to those of hLf (*Redwan et al., 2016*). It was found that camel lactoferrin has significant antibacterial activity and synergistic effects with antibiotics against MRSA. We also confirmed the superiority of cLf antibacterial activities (*Redwan et al., 2016*). Recently, we analyzed the synergistic killing of pathogenic *Escherichia coli* using camel lactoferrin from different clans of Saudi camels and various antibiotics and found that cLf has the superior potential killing of *E. coli* over bLf and hLf, and this potential can be further synergistically enhanced if cLf is combined with antibiotics (*Almehdar et al., 2019*). We also reported bacteriostatic and bactericidal activities of cLf against *Salmonella enterica* Serovar Typhi (*Almehdar et al., 2020*).

In this study, we demonstrate potential bacteriostatic and bactericidal mechanisms by which camel lactoferrin can kill *Salmonella enterica* Serovar *typhimurium* LT2 and *Shigella sonnei* ATCC 25931. Furthermore, we compared its antibacterial activity to hLf or bLf. Previously, human milk lactoferrin was found to be bactericidal for *S. typhimurium* and *E. coli via* bacteriostatic and bactericidal effects and was suggested to perform bacterial killing by direct contact; direct LPS-binding by Lf (*Ellison III & Giehl, 1991*). Additionally, lactoferrin and transferrin were proved to cause damage of *S. typhimurium* SL696 and *E. coli* CL99-2 outer membrane besides releasing their LPS, and both of these effects were modulated by $Ca^{2+}$ and $Mg^{2+}$ (*Ellison III et al., 1990*). *El Agamy et al. (1992)* reported the *in vitro* antibacterial effects of cLf against *Salmonella typhimurium* and found that the average sizes of cLf inhibitory zones of bacterial growth were 17.4–18.2 mm. Human lactoferrin was also found to impair the capability of *S. sonnei, Shigella flexneri,* and *Shigella dysenteriae* to adhere and invade HeLa cells *via* affecting key virulence proteins responsible for uptake of bacteria by host cells (*Willer Eda, Lima Rde & Giugliano, 2004*). Likewise, several studies have demonstrated the anti-inflammatory activities of both recombinant hLf and native bLf tested in models of bacterial infection by *Salmonella enterica* serovar Typhimurium in susceptible BALB/c mice (*Drago-Serrano et al., 2010*; *Mosquito et al., 2010*; *Drago-Serrano*

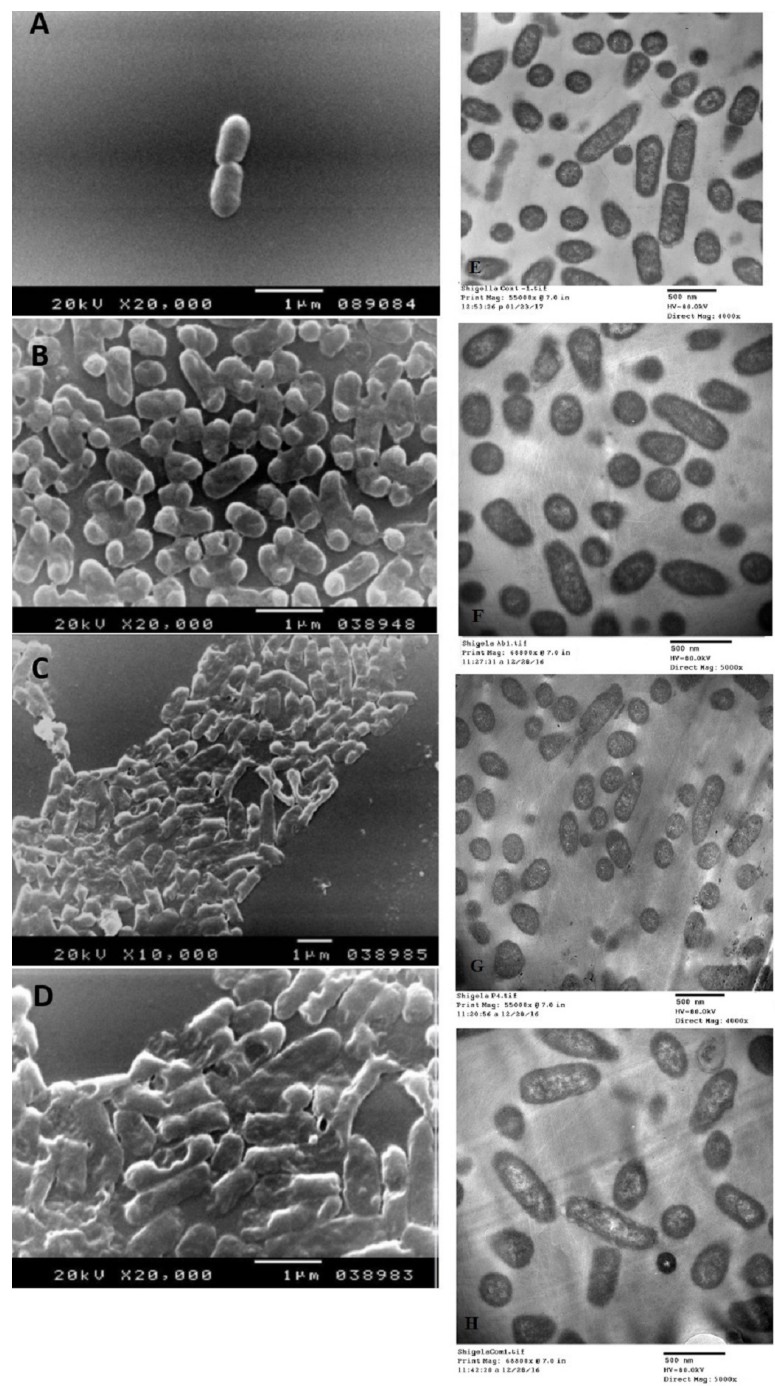

**Figure 7 Scanning electron microscope images (A–D) and transmission electron microscope images (E–H).** (A) Untreated *S. sonnei*; (B) Antibiotic (carbenicillin)-treated *S. sonnei*; (C) cLf-treated *S. sonnei*; (D) Synergistic effects of cLf-carbenicillin treatment on *S. sonnei*. The bacterium lost its viability and membrane organization and is completely destructed; (E) Untreated *S. sonnei*; (F) Antibiotic (carbenicillin)-treated *S. sonnei*; (G) cLf-treated *S. sonnei*. (continued on next page...)

**Figure 7 (…continued)**
Black dots (gold nanoparticles) scattered extracellularly and in intracellular compartments besides the destructive effects of Lf (bacterial membrane lost its organization and seems as eroded). (H) Combined effect of cLf-carbenicillin on *S. sonnei*. Black gold particles are scattered extracellularly and in intracellular compartments in addition to the vigorous destructive and membrane fragmentation effects. The bacterial membrane, which lost its organization, seems as eroded. Cells cytoplasm seems empty in white areas.

*et al., 2017*). Results of these studies suggested that Lf treatments favored the resolution of infection and protected mice from tissue damage caused by intestinal inflammation. Involved mechanisms were reported to be the increase in the production of antibodies and down-modulation of intestinal inflammation.

Our results obtained from various microbiological and protein analysis techniques revealed three significant facts, the superiority of cLfs to hLf or bLf, the similarity in the antibacterial activities of cLfs from different camel clans, and synergism between Lfs and antibiotics in inhibitory activities against *S. typhimurium* and *S. sonnei* growth. These results agree with previous reports, which revealed that cLf was the most active antibacterial agent among other lactoferrins against many pathogens (*Redwan et al., 2016*; *Almehdar et al., 2019*; *Almehdar et al., 2020*; *Conesa et al., 2008*). Additionally, synergism between lactoferrin or generally iron chelators and antibiotics in antimicrobial activity was formerly proven (*Redwan et al., 2016*; *Abuelgasim et al., 2018*; *Almehdar et al., 2020*; *Luo et al., 2014*).

The established mechanisms of *in vitro* antibacterial activity of lactoferrins in this work include: impairing the integrity of the bacterial cells and their membranes that results in the release of cell proteins, affecting bacterial protein profile, the interaction of various proteins (with different molecular weights) on the bacterial membranes with cLf-biotin, bactericidal effect *via* biotinylated lactoferrins-bacterial LPS binding, causing morphological changes both extracellularly and intracellularly after *S. typhimurium* and *S. sonnei* treatment by carbenicillin or cLf alone and in combination as demonstrated by electron microscopy, and finally exhibiting bacteriostatic effect based on Lfs ability to bind free iron. We proved all of these mechanisms altogether against *S. typhimurium* and *S. sonnei* for the first time in this work and in previous studies on the Lf action against MRSA, *E. coli* , and *Salmonella enterica* serovar Typhi (*Redwan et al., 2016*; *Almehdar et al., 2019*; *Almehdar et al., 2020*). However, Lf-LPS binding was demonstrated previously in several studies (*Ellison III & Giehl, 1991*; *Appelmelk et al., 1994*; *Drago-Serrano et al., 2012*).

The ability of lactoferrins to chelate free iron, which is vital for the growth of bacteria is responsible for their bacteriostatic effect (*Jahani, Shakiba & Jahani, 2015*). Generally, deficiency of iron prevents bacterial growth, prevents *in vitro* formation of bacterial biofilm, and forces bacteria to move; thus, preventing their adherance to surfaces (*Jahani, Shakiba & Jahani, 2015*; *Legr et al., 2005*; *Arnold, Brewer & Gauthier, 1980*). However, this bacteriostatic effect can be reversed by iron supplementation as observed in this study, our previous work (*Redwan et al., 2016*; *Almehdar et al., 2019*; *Almehdar et al., 2020*), and earlier reports (*Jahani, Shakiba & Jahani, 2015*; *Arnold, Brewer & Gauthier, 1980*; *Kell, Heyden & Pretorius, 2020*; *Mattar et al., 2021*). In this study, it was found that MIC values obtained for cLf and hLf or bLf against *S. typhimurium* and *S. sonnei* after iron addition inceased by eight times and four times, respectively compared to their MIC values without

iron addition. Also, iron saturation of Lf can reverse this bacteriostatic effect *in vivo* (*Jahani, Shakiba & Jahani, 2015*; *Kell, Heyden & Pretorius, 2020*). Our results showing cLf binding to bacterial membrane proteins of *S. typhimurium* and *S. sonnei* also support and agree with the previous reports on different bacterial strains, which demonstrated the presence of multiple and different lactoferrin-binding bacterial membrane proteins (*Drago-Serrano et al., 2010*; *Drago-Serrano et al., 2017*; *Schryvers & Morris, 1988*; *Staggs et al., 1994*; *Dhaenens, Szczebara & Husson, 1997*; *Tomita et al., 1998*; *Fang & Oliver, 1999*; *Park, Almeida & Oliver, 2002*; *Yu & Schryvers, 2002*; *Ochoa & Cleary, 2009*; *Rahman et al., 2009*; *Beddek & Schryvers, 2010*; *Morgenthau et al., 2014*; *Samaniego-Barrón et al., 2016*).

The inhibitory activities of cLf against *S. typhimurium* and *S. sonnei* are in agreement with our previous reports on other enterobacteriaceae (*E. coli*, *S. typhi*) and MRSA (*Redwan et al., 2016*; *Almehdar et al., 2019*; *Almehdar et al., 2020*). Even though the bactericidal activity of cLf was investigated in several earlier studies (*El Agamy et al., 1992*; *Conesa et al., 2008*; *Caccavo et al., 2002*; *Badr et al., 2017*), its antibacterial activity against *S. sonnei* was not analyzed in those reports, and against *S. typhimurium* was studied only by *El Agamy et al. (1992)*. Moreover, although these reports revealed the presence of the inhibitory activity of cLf, no MIC data, monitoring bacterial growth, time-kill assay, examining cLf binding to bacterial membrane proteins, or detailed analysis of synergistic antibacterial action were reported. Therefore, this study may be the first report, where the antibacterial efficiency of cLf against *S. typhimurium* and *S. sonnei* was exhaustively evaluated in comparison to hLf and bLf, and all potential bacteriostatic and bactericidal mechanisms by which camel lactoferrin can kill *Salmonella enterica* Serovar *typhimurium* LT2 and *Shigella sonnei* were demonstrated. Of note that, more recently the Lf activity was explored against the current pandemic agent SARS-CoV-2 *in vitro* and *in vivo*, which demonstrated promising inhibitory effects in free and capsulated form as reviewed (*Mattar et al., 2021*; *Gallo et al., 2022*; *Wotring et al., 2022*).

The results obtained in this work suggest the need for the examination of the potential use of cLf alone or in combination with antibiotics in preclinical and clinical trials as orally administered treatment to *S. typhimurium* infection and shigellosis. Our findings combined with the results obtained by *Willer Eda, Lima Rde & Giugliano (2004)* also suggest that camel milk may act by inhibiting adhesion of bacteria to host cells, invasion of host cells, and growth of *S. sonnei*, therefore, preventing shigellosis.

## CONCLUSIONS

The present study confirmed that cLf, bLf, and hLf have the potential to efficiently kill *S. typhimurium* and *S. sonnei in vitro*. Camel lactoferrin was demonstrated to have superior and differential potential to kill both pathogens as compared with bLf and hLf. On the other hand, cLf subtypes purified from milk of four clans of Saudi camels have similar antibacterial effects. Furthermore, our analyses confirmed and supported four different previously nominated mechanisms by which cLf killed *S. typhimurium* and *S. sonnei*: iron chelation; induction of the release, appearance, disappearance, or high expression of some bacterial proteins; binding to the bacterial LPS and multiple membrane proteins; and

impairing the integrity of the bacterial cells and their membranes. However, this work has some limitations, and further investigations are needed to establish and characterize the cLf regions possessing high binding affinity to bacterial membrane proteins and/or LPS and to identify these targeted molecules. Additionally, focused research is needed to understand if these regions are similar or different from those of bLf or hLf. Finally, the superior antibacterial activity of cLf should be confirmed *in vivo*.

### Funding

This work was supported by the King Abdulaziz City for Science and Technology General Directorate of Research Grants Programs, under grant no. LGP-35-84. The funders had no role in study design, data collection and analysis, decision to publish, or preparation of the manuscript.

### Grant Disclosures

The following grant information was disclosed by the authors:
King Abdulaziz City for Science and Technology General Directorate of Research Grants Programs:  LGP-35-84.

### Competing Interests

Vladimir N. Uversky is an Academic Editor for PeerJ. The other authors declare that they have no competing interests.

### Author Contributions

- Hussein A. Almehdar conceived and designed the experiments, analyzed the data, authored or reviewed drafts of the article, and approved the final draft.
- Nawal Abd El-Baky performed the experiments, analyzed the data, prepared figures and/or tables, authored or reviewed drafts of the article, and approved the final draft.
- Ehab H. Mattar performed the experiments, analyzed the data, prepared figures and/or tables, authored or reviewed drafts of the article, and approved the final draft.
- Raed Albiheyri performed the experiments, analyzed the data, authored or reviewed drafts of the article, and approved the final draft.
- Atif Bamagoos performed the experiments, analyzed the data, authored or reviewed drafts of the article, and approved the final draft.
- Abdullah Aljaddawi performed the experiments, analyzed the data, authored or reviewed drafts of the article, and approved the final draft.
- Vladimir N. Uversky analyzed the data, authored or reviewed drafts of the article, and approved the final draft.
- Elrashdy M. Redwan conceived and designed the experiments, performed the experiments, analyzed the data, prepared figures and/or tables, authored or reviewed drafts of the article, and approved the final draft.

## Data Availability

The data are available in the article and Supplementary File.

## Supplemental Information

Supplemental information for this article can be found online at http://dx.doi.org/10.7717/peerj.14809#supplemental-information.

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
