# Peer review of "Exploring the mechanisms by which camel lactoferrin can kill Salmonella enterica serovar typhimurium and Shigella sonnei"

_PeerJ, doi:10.7717/peerj.14809_

## Round 0.1 · original submission · Minor Revisions

Please modify the manuscript as per reviewer's comments.

Reviewer 1 ·

Basic reporting

1. The abstract contained overly detailed experimental results but lacked a brief background of the field and knowledge gaps. In addition, it is hard to tell apart the existing knowledge and the current findings. I suggest structuring the abstract as follows: introduction - knowledge gap - short description of the approach – 1~2 sentences on the results – the implication of the findings.
2. The results section contained substantial observations but was poorly organized and difficult to read. I suggest starting with the phenotype (growth inhibition, killing, etc.) and following up with mechanisms.
3. Statistical significance should be marked on the figures, and the groups compared should be included in the figure legends of Figures 1, 2, and 4.
4. In lines 273-276, what message did the authors want to deliver by showing antimicrobial resistance to antibiotics? And similarly, in lines 314-317, what does the reverse of the inhibitory activity mean?
5. The time-kill assay reported in Table 7 is better suited with a bar/line graph.
6. In lines 242-247, the authors described the content of Figure 2 and Table 1, which should be part of the legends and not the results. Similarly, the sentences in lines 260-263 should be part of the methods, not the results section. And the description in lines 322-328 should be moved to methods.
7. The use of language is sometimes vague and unclear in the manuscript. Please rewrite the following phrases: the sentence in lines 53-55 and 'personal communication' in line 57. In lines 94-96, it is unclear how cLf can kill bacteria via bacteriostatic mechanisms. Bacteriostatic molecules, by definition, inhibit growth without lowering CFU.

Experimental design

1. In Figure 4, the authors omitted two critical negative controls: 1 lactoferrin from other sources and 2. Cytosolic protein from S. Typhimurium and S. sonnei to demonstrate a specific interaction between BMP and cLf instead of general 'stickiness.'
2. In Figure 5G, the authors wrote, "Most of the bacterial cells are stunted, lost their general rod shape, and became rounded.", but the images shown didn't support this claim. Since these are TEM images, depending on how the bacteria were sliced during microscopy, bacteria spliced cross-sectionally will naturally look round, but that shouldn't be taken as their loss of shape. TEM images of untreated bacteria are needed for comparisons.

Validity of the findings

1. In line 236, the authors' claim that cLf isolated from camel milk 'have no structural differences in their glycosylation moieties' is not supported by the evidence shown in Figure 1, which contained no structural information.
2. I am not convinced by the comments on cell morphology with the SEM and TEM images shown in Figures 5 and 6. Since the samples were sliced before staining, gold particles will be inside the cell with and without membrane damage. In addition, only one view is shown for each bacterium + treatment, and the images (especially Figure 5) contained very few bacteria.

Additional comments

In this manuscript, Almehdar et al. studied the mechanism by which camel lactoferrin exhibits antimicrobial activities on S. Typhimurium and S. sonnei. The authors combined various approaches and demonstrated the scientific basis for the therapeutic value of camel milk. However, the conclusions cannot be fully supported by experimental evidence, some critical controls were missing, and the writing needs to be improved.

Reviewer 2 ·

Basic reporting

In this study, Hussein A. Almehdar et al investigated cLf as a possible theraputic option against highly transmissible strains of Shigella sonnei and Salmonella typhimurium. Camel lactoferrin has been shown to inhibit the growth of both bacterial strains at concentrations ranging from 0.125 to 3 mg/ml. cLf has been shown to interact with LPS and also change the protein expression profile of both the pathogens.

Experimental design

No Comment

Validity of the findings

No commen

Additional comments

• It is too early to conclude that there is no structural variation in the glycosylation moieties involved in the interaction between LPS and Lactoferrin based on no difference in binding with LPS for cLf subtypes in the ELISA.
• Out of 4 clf which one was used in figure 2 experiment? Statistical analysis in figure 2 has not been shown.
• The western blot image of cLf- biotin interaction with bacterial membrane proteins on a western blot has not been shown. What was the negative control used in the experiment showing interaction of membrane proteins of Shigella sonnei and Salmonella typhimurium with cLf-biotin
• Is the iron binding capacity of cLf different from the other Lactoferins?

·

Basic reporting

1. The authors write in abstract “Exploring molecular mechanisms by which camel lactoferrin can kill S. typhimurium and S. sonnei revealed that cLf affects expression of various bacterial proteins. There is no evidence for the same in the article. “Besides, it interacts with bacterial lipopolysaccharides (LPS) and numerous membrane proteins of S. typhimurium and S. sonnei, with each bacterial strain
posessing distinctive binding membrane proteins for lactoferrin” The abstract has been overstated with clearly no scientific evidence for same.

2. The authors report in abstract that cLf can kill S. typhimurium and S. sonnei by four molecular mechanisms, such as iron chelation, induction of
the release, appearance, disappearance, or high expression of some bacterial proteins, binding to bacterial LPS and membrane proteins, and impairing the integrity of the bacterial cells and their membranes. However, the authors have not shown any evidence for the same. Although there are reports that show possible mechanisms of action of lactoferrin by sequestering iron or binding with Lipid A portion of LPS but the authors don’t provide the evidence of the same. The authors need to rewrite the abstract mentioning or putting together their specific observations and conclusions they have drawn from later.

3. Overall, the image representation is not upto the standard of journals. The authors prefer tables and values over graphs and image representations. The authors also use different fonts in the paper that should be changed to just one font. The overstatement in the abstract should be reduced to the author's own findings.

Experimental design

1-Line 231-237: The authors show by ELISA that cLF was more reactive than bLf and hLf against S. typhimurium or S. sonnei LPS, with no evident differences between the cLf sub-types. The authors have shown the same by indirect ELISA. However the authors could have performed a competitive ELISA which could have shown better that cLF is more reactive than hLF and bLF. The authors overstate their observation by saying that 4 different cLf sub-types have no structural differences in their glycosylation moieties that are involved in the interaction between LPS and lactoferrin giving no scientific evidence for later. LPS is an amphipathic molecule with an overall negative net charge due to the negatively charged phosphoryl groups of LPS to which proteins and peptides with an exposed positively charged domain could interact via electrostatic forces. Hydrophobic interactions involving the fatty acid residues of lipid A and hydrophobic amino acids have also been postulated to participate in the mechanism of LPS binding.


2. The authors state the interaction of BMP with LF using western blotting, however the authors fail to provide an evidence or western blot image for the same. Line 247-248 Protein profile of S. typhimurium or S. sonnei cell clearly changed on SDS-PAGE over time (from 30 min to 150 min) depending on the used antibacterial agents in treatment (no image representation of the same).

3. Line 267-270 The authors show the interaction of BMP crude extract of S. typhimurium or S. sonnei exposed to cLf-biotin via ELISA. The authors exaggerate the data by quoting “we propose that cLf could kill S. typhimurium and S. sonnei via a bactericidal mechanism that involves binding to bacterial LPS and membrane proteins”, however providing no evidence for the same. A mere interaction is not proof of a bactericidal mechanism.

4. Line 308-310 The authors performed MIC of antibacterial agents by broth microdilution assay. The authors have not mentioned the concentration range they used for LF. Also, the authors can represent the same in a bar graph showing higher inhibitory concentration for other LFs compared to cLF.

5-Line 357-359 The author shows membrane distortion of bacteria by TEM image while the gold nanoparticles black dots are not clear into the image. I would appreciate it if the authors could provide a good quality image with high magnification that should show the membrane distortion. Also, The control image in TEM analysis is missing.

Validity of the findings

The authors have previously demonstrated that the purified camel lactoferrins from different Saudi camel clans, as well as human and bovine lactoferrins possess antimicrobial properties against Salmonella enterica serovar Typhi. The authors further showed that all cLfs showed superior antibacterial potentials in comparison to hLf or bLf. In this paper, the authors aim at exploring/understanding the molecular mechanisms by which camel lactoferrin can kill S. typhimurium and S. sonnei. For this, the authors show that the camel lactoferrin interacts with lipopolysaccharides and membrane proteins of S. typhimurium and S. sonnei inducing extracellular and intracellular morphological changes leading to antimicrobial action by lactoferrin. The reviewer believes the article requires some major modifications.

---

## Round 0.2 · accepted · Accept

As a Section Editor, I have been asked to take over handling this submission after the original Academic Editor invited a large number of reviewers after the initial minor revisions decision.

The initial reviewers are pleased with the improvements to the manuscript and it is ready to publish, congratulations.

Reviewer 2 ·

Basic reporting

In this study, Hussein A. Almehdar et al investigated cLf as a possible theraputic option against highly transmissible strains of Shigella sonnei and Salmonella typhimurium. Camel lactoferrin has been shown to inhibit the growth of both bacterial strains. cLf has been shown to interact with LPS and also change the protein expression profile of both the pathogens. The study supports the four different previously reported mechanisms by which cLf kills S. typhimurium and S. sonnei: iron chelation; induction of some bacterial proteins; binding to the bacterial LPS and multiple membrane proteins; and impairing the integrity of the bacterial cells and their membranes. Author has addressed the reviews queries and manuscript has been updated accordingly. My opinion is that the manuscript is ready for publication.

Experimental design

NA

Validity of the findings

NA

Additional comments

NA

·

Basic reporting

No comment

Experimental design

No comment

Validity of the findings

No comment

Additional comments

No comment

Reviewer 4 ·

Basic reporting

For figure 1, I request the authors to add error bars. It is not clear how many experimental replicates were done to generate the data presented in figure 1.

Experimental design

No Comment

Validity of the findings

No Comment

Additional comments

The authors have addressed all the comments.

·

Basic reporting

The manuscript entitled “Exploring the mechanisms by which camel lactoferrin can kill Salmonella enterica serovar typhimurium and Shigella sonnei” investigated potential role of camel lactoferrin as an antimicrobial agent especially against of Salmonella enterica Serovar typhimurium and Shigella sonnei. Despite some limitations and shortcomings in experimental design, these findings provide evidence that camel lactoferrin (cLf) kills Salmonella enterica Serovar typhimurium and Shigella sonnei more effectively than as compared to bLf or hLf.

1. There is no description in the introduction about Lfs role in host nutritional immunity and host-pathogen interactions. Introduction needs some background information regarding the proven roles of Lfs in host immune response and antimicrobial activity in response to bacterial infections.

2. More explanation is needed in the discussion about four different mechanisms by which cLf killed S. typhimurium and S. sonnei. And it would be nice if authors include a conceptual model of the 4 mechanisms by which cLF kills S. typhimurium and S. sonnei.

Experimental design

1. Authors reported that supplementation of Iron reverses the inhibitory activity cLf (in the case of MIC determination experiments). As antibacterial activity of combined cLF and antibiotics is greater than cLf alone. Why did not authors include Time kill assay with iron supplementation to see if iron supplementation could decrease susceptibility to cLf alone or in combination with antibiotics. Authors did not provide explanation for possible mechanisms that renders bacteria more susceptible to combined treatment of cLf and antibiotics, just simply stated it is formally proven (line 403). Is it because of cLf restricting the bacterial access to the iron, lacking access to vital nutrient for bacterial physiology as an essential component of metabolic enzymes and regulatory proteins making bacteria more susceptible to antibiotics? Or it is because of cLf modulating bacterial proteins expression or directly binding to bacterial LPS or bacterial surface proteins making bacteria more susceptible to antibiotics. To understand possible mechanisms behind the synergistic antimicrobial activity of cLf and antibiotics, Time kill assay with iron supplementation is required.

2. I don’t understand why Figure 4 is included in the main figures and what it is explaining. Authors just showed protein profiles of the whole cell lysate and membrane protein profiles and claiming (line 339) that profile of membrane proteins was found to be different from the profile of whole-cell lysate. That is obvious when comparing proteins from different cellular compartments. In the figure 4 authors should include the BMP extract and proteins of whole cell lysate treated with cLf for comparison. In the supplemented material I don’t find the kinetic proteins profiles of untreated bacteria. How did you compare if you don’t have protein profiles of untreated. 0 mins means untreated or immediately after adding cLf or antibiotic or both? Please clarify.

3. Since the bacterial cell integrity of the bacterial cells and their membranes is compromised when treated with Lf alone or in combination with antibiotics, it could be due to the changes in bacterial cell wall proteome composition. Since authors observed changes in protein profiles, It would be better to have mass spectrometry analysis that would provide information about the specific proteins that were altered and how that proteins are linked to the observed phenotypes.

4. It would be better if (possible) authors perform pulldown assay with cLf biotin using streptavidin Bead and mass spectrometry to identify bacterial proteins interacting with cLf or any other pulldown assay.

Validity of the findings

1. In figure 5 there is no negative control (BMP coated well without adding cLF-biotin) to ensure there is no unspecific binding of streptavidin-peroxidase to BMP extract of S.typhimurium or S. sonnei. What is the control cLf means, cLf without biotin tag added to BMP extract coated wells? Or wells coated with cLf alone? Please clarify.

2. Why there is no western blot picture, only table with detected proteins and their molecular weight. Is it possible for authors to predicts what are those proteins based on the molecular weight?

Reviewer 6 ·

Basic reporting

No comments

Experimental design

No comments

Validity of the findings

No comments

Additional comments

The revised manuscript can be accepted for publication.

·

Basic reporting

No comment

Experimental design

No comment

Validity of the findings

No comment

Additional comments

The manuscript “Exploring the mechanisms by which camel lactoferrin can kill Salmonella enterica serovar typhimurium and Shigella sonnei” by Almehdar et al., addresses an important and timely topic as there is a severe concern over the excessive use of antibiotics and the generation of new antibiotic resistant strains of bacteria rapidly. Overall, the study seems well conducted and very well written. The results are well organized, and the methods are explained in reproducible manner. Specific comments are mentioned below.

1. Introduction section- Line 74-76- “Therefore, safe and effective unconventional therapy is strongly recommended for managing these multi-drugs resistant and highly transmissible strains of S. sonnei and S. typhimurium” Authors should, in very brief, add some of these unconventional therapies that are being used against multi-drugs resistance other that camel milk.

2. Line 261-263- Furthermore, the data presented in Table S1 demonstrate that the……… Lfs not only inhibit bacterial growth but also induce cell death via lysis- Did the authors check the cell death using any markers of cell death or lysis? If not, this statement is not true as the authors are not sure if the death of the bacteria is indeed due to cell death or some other reason.

3. Figure 4- The SDS blots panel looks a bit blurry, replace with better images with clear bands, it’s difficult to differentiate between protein profiles with these images.

4. Figure 6- The specifications written below all the images in the right-side panel is not clear and impossible to read. Replace with clear images. Do the same in Figure 7.

5. There seems to be a repetition of all the Tables before and after the Figures, make changes accordingly.

Reviewer 8 ·

Basic reporting

Manuscript is well written. The results section is adequately explained.

Experimental design

No comments

Validity of the findings

- Authors show the change in the expression of certain proteins based on their molecular weight. But its not right to correlate them with being involved in cLF-Bacterial interaction or regulating their growth or pathogenicity. Authors should make this point clear in discussion.

Additional comments

- Show the significant diff in Fig 2.

- What is the significance of showing Fig 4. Mention in the result section

- Authors should show the transblot image along with Table 6.

- What about the interaction of bLF and hLF in Fig 5. Mention which cLF (1-4) was used.

- Is there any information available on the effect of endotoxin release by cLF treatment (compared to antibiotic treatment), add that to discussion.
- Did authors take any effort to test the in-vivo activity of cLF in mouse infected with Salmonella or other bacterial pathogens?